# *Folliculin* (*FLCN*) in Thyroid Tumors: Incidence, Significance, and Role as a Driver Gene and Secondary Alteration

**DOI:** 10.3390/curroncol32040224

**Published:** 2025-04-11

**Authors:** Faisal A. Hassan, Camryn Slone, Robert J. McDonald, Julie C. Dueber, Adeel M. Ashraf, Melina J. Windon, Oliver J. Fackelmayer, Cortney Y. Lee, Therese J. Bocklage, Derek B. Allison

**Affiliations:** 1Department of Pathology & Laboratory Medicine, University of Kentucky College of Medicine, Lexington, KY 40506, USA; 2Kentucky College of Medicine, University of Kentucky, Lexington, KY 40506, USA; 3Department of Otolaryngology—Head and Neck Surgery, University of Kentucky College of Medicine, Lexington, KY 40506, USA; 4Department of Surgery, Division of Endocrine Surgery, University of Kentucky College of Medicine, Lexington, KY 40506, USA; 5Markey Cancer Center, Lexington, KY 40536, USA

**Keywords:** thyroid cancer, molecular testing, Brit–Hogg–Dube, folliculin, preoperative molecular testing

## Abstract

Thyroid carcinomas are driven by diverse molecular alterations, but the tumor suppressor gene folliculin (*FLCN*), best known for its role in Birt–Hogg–Dubé (BHD) syndrome, has received limited attention in thyroid tumors. Here, we describe two thyroid tumors with pathogenic *FLCN* alterations—one germline and one somatic—and analyze the broader prevalence and significance of *FLCN* in thyroid carcinomas using multiple large sequencing datasets, including ORIEN-AVATAR. Patient 1, with a germline *FLCN* mutation and a history of BHD syndrome, presented with a well-circumscribed oncocytic adenoma. Molecular testing confirmed biallelic *FLCN* inactivation, but no additional mutations or aggressive features were observed, and the patient remained disease-free post-thyroidectomy. Patient 2 harbored a somatic *FLCN* mutation in an oncocytic poorly differentiated thyroid carcinoma, which exhibited extensive angioinvasion, high proliferative activity, and concurrent TP53 and RB1 mutations. The tumor progressed with metastatic disease despite multimodal treatment. Thyroid carcinomas revealed *FLCN* alterations in 1.1% of cases. Pathogenic mutations were rare but associated with oncocytic morphology, while homozygous deletions occurred more frequently in genomically unstable tumors, including anaplastic thyroid carcinoma. These findings suggest *FLCN* mutations may act as early oncogenic drivers in oncocytic thyroid neoplasms, while deletions represent secondary events in aggressive tumor evolution. The lack of *FLCN* coverage in standard thyroid molecular panels likely underestimates its clinical relevance. Including *FLCN* in genetic testing could improve tumor detection and characterization, particularly in BHD patients who may benefit from routine thyroid screening. Further studies are needed to clarify *FLCN*’s role in thyroid cancer pathogenesis.

## 1. Introduction

Thyroid carcinomas represent a diverse group of malignant tumors, often driven by specific molecular genetic alterations that correlate with distinct histologic features and clinical behavior [1]. As increasing numbers of thyroid tumors are now being sequenced for preoperative clinical decision-making, diagnostic characterization, and therapeutic considerations, new genetic alterations continue to emerge [2]. Some thyroid carcinomas are associated with germline mutations, as observed in cribriform morular carcinoma and subsets of medullary and papillary thyroid carcinomas [3,4,5]. Thyroid nodules have also been reported in patients with Birt–Hogg–Dubé (BHD) syndrome, an autosomal dominant condition caused by a germline *folliculin* (*FLCN*) mutation [6,7,8]. Additionally, somatic *FLCN* alterations, distinct from germline mutations, have been reported in tumors at other sites [9]. However, published details regarding both germline and somatic *FLCN* alterations in thyroid nodules remain sparse.

*FLCN*, located on chromosome 17p11.2, encodes the folliculin protein, a tumor suppressor that regulates key cellular processes, including mTOR and AMPK signaling pathways, which control cell proliferation, metabolism, autophagy, and mitochondrial function [10,11,12]. While BHD syndrome is frequently associated with cutaneous fibrofolliculomas, oncocytic renal tumors, and lung cysts, an increasing number of somatic *FLCN* alterations have been identified in a range of tumors, including parathyroid adenomas and carcinomas, salivary gland oncocytomas, lung adenocarcinomas, basal-like breast cancers, malignant PEComas, and colorectal adenocarcinomas, among others [8,9,13,14,15,16,17,18,19,20]. *FLCN* mutations disrupt cellular homeostasis by impairing mitochondrial biogenesis and metabolism, leading to diverse histologic and clinical phenotypes depending on the tumor type and context [7,13,14,15,16]. Despite these insights, the role of *FLCN* in thyroid neoplasms remains largely unexplored, with only a few isolated reports linking *FLCN* mutations to thyroid tumors [6,8,21].

In this report, we describe two rare thyroid tumors with pathogenic *FLCN* alterations: one harboring a germline (heriditary) *FLCN* mutation in a patient with BHD syndrome and the other with a somatic (acquired) *FLCN* mutation. To further investigate the incidence and clinical relevance of *FLCN* alterations in thyroid carcinomas, we analyzed data from multiple large publicly available sequencing databases, as well as the ORIEN-AVATAR dataset. By characterizing the clinical, histologic, and genetic profiles of these two *FLCN*-mutated thyroid neoplasms and examining the broader role of *FLCN* alterations in thyroid carcinoma pathogenesis, we seek to expand the understanding of the genetic landscape of thyroid tumors.

## 2. Materials and Methods

This study was conducted in two phases, which included two index patients encountered during routine clinical practice, followed by an analysis from multiple large sequencing datasets.

### 2.1. Index Patients

Two patients were identified at our institution, each presenting with thyroid tumors that were found to harbor pathogenic *FLCN* mutations through clinical next-generation sequencing (NGS). For each patient, we conducted a thorough review of all available clinical and laboratory data, including molecular sequencing results from the Afirma^®^ Genomic Sequencing Classifier, performed on aspirated thyroid samples, as well as MI Profile Testing through Caris^®^ Life Sciences, conducted on formalin-fixed, paraffin-embedded tissue [22,23].

Histologic slides were re-examined by four surgical pathologists with subspecialty expertise in head and neck pathology. All immunohistochemical staining was performed in a CLIA-certified, CAP-accredited laboratory, adhering to specific antibody protocols, with confirmation of appropriate reactivity using both positive and negative controls. Clinical follow-up was obtained for both patients.

### 2.2. Database Sources and Case Selection

The cBioPortal bioanalytical platform was employed to analyze thyroid carcinoma sequencing data from both the Orien-Avatar database and several publicly available datasets [24,25]. The cohort was generated by identifying cases with *FLCN* gene alterations, specifically focusing on mutational and copy number alteration (CNA) status. *FLCN* alterations that were determined to be likely oncogenic or oncogenic based on data found in the ClinVar database were identified and selected for inclusion in this study. For CNA analysis, this meant that only cases with deep deletion/biallelic loss were determined to be oncogenic. Additional likely oncogenic and oncogenic mutations for each case were also recorded according to data found in the OncoKB database.

### 2.3. Orien-Avatar Dataset

For the Orien-Avatar cohort, both germline and somatic sequencing data were available and analyzed. All cases with *FLCN* alterations meeting the above criteria were included. Clinical and pathological data were collected for each case with *FLCN* alterations. The following variables were recorded: patient age at diagnosis, sex, pathologic diagnosis, somatic vs. germline alteration status, the specific *FLCN* alteration and corresponding variant allele frequency (VAF) with the number of variant and reference reads, other oncogenic or likely oncogenic mutations, family cancer history, overall survival, vital status, T stage at diagnosis, N stage at diagnosis, and presence of metastases when available.

### 2.4. Publicly Available Datasets

In addition to the Orien-Avatar data, publicly available datasets were included in the analysis. These datasets were analyzed on the cBioPortal platform and included thyroid carcinoma cases from the GATCI 2024 Cell Reports study, the MSK 2016 JCI study, and the TCGA datasets [26,27,28]. As with the Orien-Avatar cohort, only cases with *FLCN* alterations meeting the above criteria were included. Sequencing data from these studies were derived from somatic tumor assessment only. For all cases, the following clinical and pathological variables were recorded when available: age range at diagnosis, sex, pathologic diagnosis, the specific *FLCN* alteration, VAF with the number of variant and reference reads, other oncogenic or likely oncogenic mutations, overall survival, vital status, T stage at diagnosis, and presence of metastasis. The log-rank test was employed to calculate the *p*-value and evaluate overall survival differences.

## 3. Results

### 3.1. Patient 1

#### Clinicopathologic Findings

A 44-year-old female with BHD syndrome (confirmed via *FLCN* germline mutation) and a history of multifocal chromophobe renal-cell carcinoma presented with a newly detected thyroid nodule on routine physical exam. A thyroid ultrasound revealed a 2.4 cm TI-RADS 5 nodule in the left thyroid lobe, which was hypoechoic with internal blood flow. Fine needle aspiration (FNA) of this nodule yielded a diagnosis of “suspicious for papillary thyroid carcinoma” (Bethesda category V). Afirma^®^ Genomic Sequencing Classifier (GSC) and Xpression Atlas (XA) detected no variants or fusions, with the ensemble classifier deemed non-applicable due to insufficient data on malignancy risk for Bethesda V/Afirma XA-negative cases. The patient subsequently elected for a total thyroidectomy.

Upon pathologic examination, a well-circumscribed, 2.9 cm nodule was found in the left thyroid lobe. Microscopically, the tumor demonstrated a diffuse solid growth pattern without colloid, and the cells exhibited striking oncocytic characteristics, including abundant granular eosinophilic cytoplasm, enlarged nuclei with variably prominent nucleoli, and prominent pseudo-inclusions filled with eosinophilic material (Figure 1). Some nuclei contained degenerative changes and variable amounts of nuclear clearing and nuclear contour irregularity, perhaps contributing to misclassification on FNA. Notably, mitoses were absent, as were necrosis, angioinvasion, lymphovascular invasion, and perineural invasion. The tumor was completely excised, and all nine lymph nodes examined were free of tumor involvement. On immunohistochemical (IHC) staining, tumor cells showed patchy expression of CD56, but the tumor was negative for broad-spectrum keratins, PAX8, TTF1, thyroglobulin, calcitonin, HBME-3, CEA, PTH, GATA3, p63, SOX10, and S-100. The Ki67 proliferation index was low (~1%), and a PD-L1 showed cytoplasmic and membranous staining in 30% of tumor cells.

A comparison with the patient’s previously resected renal tumors demonstrated distinct differences in histologic and immunohistochemical staining patterns, confirming that the thyroid tumor was not metastatic from her chromophobe renal-cell carcinoma. The tumor was ultimately found to be consistent with an oncocytic adenoma; however, given the unusual staining pattern and the patient’s BHD syndrome, the tumor was sent for molecular testing. Next-generation sequencing was performed by Caris^®^ Life Sciences on the tumor from the thyroid, which showed a pathogenic p.M394fs mutation (VAF 82%) in the *FLCN* gene, indicative of biallelic inactivation (double hit) and supportive of a neoplastic process. This mutation is consistent with the germline testing performed by Ambry Genetics^®^ which detected a heterozygous c.1179delC (p.M394Cfs*74) pathogenic germline mutation in the *FLCN* gene. Finaly, the tumor mutational burden (TMB) was low (2), and no abnormalities were detected in BRAF, NRAS, HRAS, NTRK1/2/3, RET, or TERT. In summary, the findings are consistent with an *FLCN*-driven oncocytic adenoma.

Nine months post-thyroidectomy, the patient has remained disease-free, with no evidence of recurrence on follow-up ultrasounds and PET scans.

### 3.2. Patient 2

#### Clinicopathologic Findings

A 59-year-old female with a long clinical history of goiter presented with compressive symptoms and a newly self-detected, quickly enlarging nodule in the central neck. Five years prior, the patient underwent FNA of a thyroid nodule, which was determined to be benign. Since then, multiple ultrasounds revealed stable thyroid nodules without significant changes. However, due to compressive symptoms, the patient was taken directly to the operative room at an outside hospital for a right thyroid lobectomy. The pathology was initially diagnosed as an oncocytic variant of a widely invasive papillary thyroid carcinoma (PTC).

Histopathologic review of the initial right thyroid lobectomy specimen at our institution revealed a 4.0 cm infiltrative tumor replacing most of the lobe. Rather than classic PTC morphology, the tumor displayed two distinct architectural patterns: solid diffuse sheets comprising 90% of the tumor and a papillary-like growth pattern accounting for the remaining 10%. Interestingly, there was an abrupt or sharp demarcation between these two architectural patterns. The tumor cells were densely oncocytic, characterized by abundant granular eosinophilic cytoplasm and round-to-ovoid nuclei with coarse chromatin and prominent macronucleoli. There was a notable lack of PTC nuclear features, such as nuclear grooves and intranuclear pseudoinclusions. In addition, extensive angioinvasion and lymphatic invasion were identified, and mitoses were observed at a rate of up to 4 per high-power field (HPF). Necrosis, perineural invasion, pleomorphism, and atypical mitoses were not observed. The findings were consistent with a diagnosis of an oncocytic poorly differentiated thyroid carcinoma (Figure 2).

The patient was then transferred to our institution for further management, where a more detailed assessment and additional surgeries were performed. These included a completion thyroidectomy and excision of a tumor thrombus in the right internal jugular vein, as well as removal of nodules in the right parapharyngeal space and strap muscle. The completion thyroidectomy revealed two small oncocytic nodules without malignant features while the jugular vein thrombus, strap muscle, and parapharyngeal space disease showed carcinoma with histologic resemblance to the primary tumor with increased mitotic activity (>10/2 mm²) and increased nuclear atypia. IHC staining of the tumor in the neck showed diffuse expression of PAX8 and TTF1, indicating thyroid origin, and negative staining for thyroglobulin, calcitonin, PTH, and GATA3, which support a diagnosis of an oncocytic poorly differentiated thyroid carcinoma. The Ki67 proliferation index was notably elevated (~30%), and PD-L1 staining demonstrated membranous reactivity in 95% of tumor cells, suggesting a high level of immune checkpoint protein expression.

To further characterize the tumor and identify potential targetable alterations, next-generation sequencing was performed by Caris^®^ Life Sciences on the tumor from the jugular vein thrombus, revealing oncogenic mutations in the following genes but no targetable alterations: *FLCN* p.E464* (VAF 85%), RB1 p.R320* (VAF 78%), and TP53 p.E336fs (VAF 80%). The tumor mutational burden (TMB) was low (3). Of note, no abnormalities were detected in BRAF, NRAS, HRAS, NTRK1/2/3, RET, or TERT.

After surgery, the patient developed metastatic disease, with lesions identified in multiple locations, including the parotid gland, mandibular bone, cavernous sinus, lung, and iliac bone. She has since undergone multimodal treatment with radioactive iodine, external beam radiation, and systemic therapies including sorafenib, pembrolizumab, and lenvatinib, with varied responses across metastatic sites over the course of two years.

### 3.3. Orien Avatar Dataset

In this dataset, 1091 patients with thyroid carcinoma were identified with *FLCN* gene coverage by NGS. Among these, 10 patients (0.92%) were found to have oncogenic *FLCN* alterations, including one patient with a germline G319fs mutation and nine patients with biallelic loss. Clinical, pathologic, and molecular details for these cases are summarized in Table 1.

Briefly, the mean age of the 10 patients was 46.1 years, with 6 of the 10 patients (60.0%) being female. Moreover, 8 of the 10 patients (80.0%) were diagnosed with papillary thyroid carcinoma, not otherwise specified (PTC, NOS), 1 patient (10.0%) had a follicular variant of PTC, and 1 patient (10.0%) with a germline mutation was diagnosed with an oncocytic carcinoma. Of these 10 patients, 8 (80.0%) had BRAF V600E mutations while 2 patients (20.0%), including the germline patient, had no likely oncogenic or oncogenic somatic mutations. Unfortunately, the patient with the germline mutation did not have *FLCN* gene coverage on their somatic tumor NGS panel, meaning the VAF is unknown. All 10 thyroid carcinoma cases exhibited copy number alterations, including amplifications and deletions at multiple chromosomal sites. At the time of the last follow-up, all 10 patients were alive, with an average follow-up time of 61.1 months. Moreover, 8 of the 10 patients (80.0%) had developed metastases. Among these, six patients (60.0%) developed metastases confined to the head, face, or neck lymph nodes. The remaining two patients (20.0%) had both head, face, or neck lymph node metastases as well as distant metastases.

### 3.4. Publicly Available Datasets

In the publicly available databases, a total of 2062 patients with thyroid carcinoma had *FLCN* gene coverage and met the inclusion criteria. Of these, 25 patients (1.21%) were identified with oncogenic *FLCN* alterations, including 1 patient with a Q220* truncating mutation and 24 patients with biallelic loss. Clinical, pathologic, and molecular details for these cases are summarized in Table 2.

Briefly, among the 24 patients with available age information, 14 patients (58.3%) were 70 years of age or older. Of the 25 patients with sex information, 14 (56.0%) were male. Twenty-four patients (96.0%) were diagnosed with anaplastic thyroid carcinoma, while one patient (4.0%) had PTC with predominantly a follicular architecture and oncocytic and focal tall cell features. Clinical follow-up data were available for 24 of the 25 patients (96.0%). Of these, 19 patients (79.2%) were deceased at the time of the last follow-up. See Figure 3 for overall survival data. Information regarding metastases was available for 22 patients (88.0%), and 15 of these patients (68.2%) had developed metastases. See Table 2 for additional details. Tumor mutational analysis was available for 24 patients (96.0%).

Canonical mutations were prevalent, with eight patients (33.3%) harboring BRAF V600E mutations and six patients (25.0%) having NRAS mutations. TP53 mutations were identified in eight patients (33.3%). One patient (4.0%) had no identified oncogenic point mutations but did have abundant copy number alterations. See Table 2 for additional details. All 25 tumors assessed for CNAs exhibited amplifications and deletions at multiple chromosomal sites, indicating significant genomic alterations. When survival data were available, overall survival was compared between patients with *FLCN* driver alterations versus those without across all public datasets included in this study. Patients with *FLCN* driver alterations exhibited significantly worse overall survival (Figure 3A). Given that 24 of the 25 patients with *FLCN* driver alterations in the public datasets included in this study had anaplastic thyroid carcinoma, a subgroup analysis was conducted to evaluate the impact of *FLCN* on overall survival in this aggressive subset (Figure 3B). While there was a trend toward poorer survival in patients with *FLCN* alterations, the difference did not reach statistical significance.

## 4. Discussion

In this study, we present the first comprehensive analysis of *FLCN* gene alterations in thyroid carcinomas, including both germline and somatic settings. Our findings reveal that *FLCN* mutations, while rare, may serve as early oncogenic drivers in thyroid neoplasia, particularly in cases exhibiting oncocytic features. Furthermore, secondary *FLCN* alterations were frequently encountered in aggressive thyroid tumors lacking oncocytic morphology, highlighting the complexity of FLCN’s role in thyroid tumorigenesis and underscoring the need for its inclusion in clinical molecular panels for thyroid cancer diagnostics.

Briefly, the FLCN protein is critical for maintaining cellular homeostasis via its interactions with folliculin-interacting proteins (FNIP1 and FNIP2) and regulation of essential metabolic pathways [13,29,30]. Its loss disrupts the balance between anabolic and catabolic processes, largely due to its role as a GTPase-activating protein for Rag GTPases involved in nutrient signaling [31,32,33]. Normally, the FLCN protein regulates the mechanistic target of rapamycin complex 1 (mTORC1) by recruiting it to lysosomes, where it controls TFE3 and TFEB—transcription factors governing lysosomal biogenesis, autophagy, and cellular metabolism [32,34,35]. When FLCN protein is absent, TFEB and TFE3 escape mTORC1 regulation, translocate to the nucleus, and activate lysosomal- and mitochondrial-related gene expression [36,37,38]. FLCN protein loss also leads to constitutive AMPK activation, enhancing catabolism and promoting metabolic adaptations that support tumor survival and growth [36,38,39].

When AMPK remains active, it drives up the expression of PGC1α, a key transcriptional coactivator that regulates mitochondrial biogenesis and oxidative metabolism [40]. Enhanced PGC1α activity increases the synthesis of mitochondrial proteins and supports oxidative phosphorylation, leading to a high density of mitochondria in the cytoplasm, producing an oncocytic appearance that is characteristic of many *FLCN*-related tumors [41,42]. This mechanism of mitochondrial accumulation is different than that observed in conventional oncocytic thyroid carcinomas. As described by Raj et al. and Gandly et al., haploidy in non-*FLCN* altered oncocytic thyroid carcinoma disrupts the balance of nuclear–mitochondrial crosstalk by reducing the dosage of nuclear-encoded mitochondrial genes [43,44]. This loss of genomic content compromises the regulation of key components of the electron transport chain (ETC). Compounding this disruption, mitochondrial DNA (mtDNA) mutations are frequently observed in these tumors, particularly in genes encoding ETC subunits, leading to impaired oxidative phosphorylation [43,44]. The combination of nuclear haploidy and mtDNA mutations amplifies mitochondrial dysfunction, creating a bioenergetic deficit that triggers compensatory mitochondrial biogenesis—producing the hallmark oncocytic phenotype. While both conventional oncocytic thyroid carcinomas and *FLCN*-mutant tumors feature mitochondrial accumulation, they differ markedly in their molecular drivers, mitochondrial function, and genomic context.

We began this study with a case of oncocytic adenoma in a BHD patient harboring a biallelic oncogenic *FLCN* mutation, with no other oncogenic alterations detected by routine RNA or DNA sequencing. This case suggests *FLCN* loss may drive oncocytic tumorigenesis, consistent with other *FLCN*-driven neoplasms. Additionally, our database review identified another BHD patient with an oncocytic carcinoma and distant metastases, also lacking other driver alterations. Together, these cases support a potential role for *FLCN* as a rare primary driver in a subset of oncocytic thyroid tumors, as seen in *FLCN*-related tumors of the kidney and parathyroid gland.

Interestingly, *FLCN* mutations were not limited to BHD patients. We also presented a patient with an oncocytic, poorly differentiated thyroid carcinoma with an oncogenic biallelic somatic *FLCN* mutation. Not surprisingly, this tumor showed co-existing *TP53* and *RB1* mutations, which along with *TERT* promoter mutations, are well known to be associated with high-grade transformation in thyroid carcinomas [27,28,45,46,47]. In addition, our database analysis revealed another patient with an *FLCN* mutation; however, this case is somewhat more complicated. This patient’s tumor was diagnosed as an anaplastic thyroid carcinoma (ATC) and contained a *BRAF* V600E mutation, in addition to *NF2* and *TP53* mutations. A lack of histologic review of this case precludes assessment of whether oncocytic features were present in any component of the tumor; however, the presence of a *BRAF* mutation would make a strong case for being a primary driver. Although, it is important to point out that the VAF of *FLCN* in this case was 54% while the VAF of *BRAF* was 43%. This finding does suggest that this *FLCN* mutation was an early alteration; however, the tumor fraction in this case is unknown, and it is unclear if this represents a monoallelic or biallelic alteration. In summary, these observations suggest that *FLCN* mutations, particularly biallelic mutations, whether occurring in BHD patients or sporadically, may contribute significantly to the pathogenesis of thyroid neoplasms that may have a propensity toward oncocytic features, underscoring the need for further study into the molecular mechanisms and clinical implications of *FLCN* alterations in thyroid cancer.

The link between BHD syndrome and thyroid tumors is less recognized, and the degree of risk remains unclear. Reported thyroid neoplasms in BHD patients include oncocytic carcinoma, medullary carcinoma, papillary carcinoma, clear-cell carcinoma, and non-invasive follicular tumors with papillary-like nuclear features—but these aggregate to very few cases [6,8,21]. Only two of these reports of thyroid tumors were examined genetically to confirm an *FLCN* alteration. The first patient was a 72-year-old white male with BHD who developed lung metastases two years after resection of his 3.5 cm, widely invasive clear-cell thyroid carcinoma [48]. The second case was reported in an abstract only with limited morphologic assessment. The patient was a 55-year-old male with BHD who presented with a 3 cm thyroid nodule and indeterminate FNA diagnosis [49]. He underwent a total thyroidectomy with lymph node dissection and was found to have an oncocytic carcinoma with nodal involvement. Oncogenic *FLCN* (H429FS*27), *TP53* (R248Q), and *DAXX* (L666fs*29) mutations were identified. Similar to patient 2 in our study, it is likely that the secondary *TP53* and *DAXX* mutations are responsible for the aggressive clinical behavior in this *FLCN*-mutated tumor. Reflecting the sparse hard data regarding thyroid tumor incidence in BHD patients, recent comprehensive reviews of familial inherited germline syndromes with a propensity for thyroid tumorigenesis excluded BHD patients [3,50]. Nonetheless, 65% of patients with BHD (18 of 22 individuals from 10 unrelated families) were found to have thyroid nodules or cysts on ultrasound examination in a French cohort [51], suggesting that the incidence of thyroid lesions in BHD may be underappreciated.

In the datasets analyzed, oncogenic *FLCN* alterations appear to be rare events in thyroid carcinoma, occurring in a total of 35 out of 3153 (1.1%) unique patients. More specifically, oncogenic mutations appear to be even rarer occurrences and represented only 2 out of the 35 (5.7%) altered cases. The remainder of the cases showed homozygous deletion, which appeared to happen in a background of frequent canonical driver mutations and significant genomic alterations (please refer to Table 1 and Table 2). When combined with our two clinical cases, it seems that *FLCN* mutations are likely to represent early alterations in a similar manner as is observed in the kidney in patients with BHD syndrome [52]. In contrast, the *FLCN* codeletions identified in this study are most likely secondary alterations. This observation is supported by two cases of PTC identified in the ORIEN AVATAR dataset which showed no *FLCN* alterations in the primary tumor but showed homozygous deletion in the metastases, which is not surprising given that the loss of *FLCN* supports tumor growth and survival. Interestingly, it is important to note that *FLCN* deletion has been known to cause clinical BHD syndrome [15]. As a result, it makes sense to include these cases in our evaluation of *FLCN*-altered thyroid tumors.

In the database analyses, it is also clear that many of these tumors with codeletion do not exhibit oncocytic features (see Table 1 and Table 2 for diagnoses), though this inference is limited by a lack of available slides for histologic review. However, many, but not all, of these tumors were ATCs, thus representing a state of dedifferentiation in a background of complex molecular alterations (see Table 1 and Table 2). Several potential explanations for a lack of oncocytic features are worth noting. One hypothesis is that these tumors undergo metabolic reprogramming toward glycolysis, known as the Warburg effect, rather than oxidative phosphorylation, a shift that reduces reliance on mitochondrial function [53,54]. Another possibility is that increased mitophagy maintains mitochondrial content within normal ranges by removing excess or damaged mitochondria [55]. Together, these mechanisms highlight the complexity of regulatory networks that could be at play.

In the follow-up data analysis, clinical behavior appeared to be more related to the underlying diagnosis and additional secondary alterations known to drive aggressive behavior, such as concurrent *TP53*, *RB1*, or *TERT* promoter mutations [45,46,56], rather than the presence or absence of an *FLCN* alteration. Though this analysis was largely biased for assessing somatic *FLCN* biallelic loss and not biallelic inactivating mutations, which do seem to occur in different overall genomic landscapes. As a result, future studies investigating the role of *FLCN* mutations as primary drivers in thyroid neoplasia, as well as their clinical significance, are warranted.

Unfortunately, standard clinical genetic commercial panels assessing thyroid lesions for malignancy will not detect an *FLCN* alteration [49,57]. This fact is underscored in patient 1, who had undergone preoperative Afirma^®^ testing with no alterations identified. If *FLCN* is the only altered gene in such a tumor, no mutation will be detected, which may imply that a non-neoplastic lesion was sampled. If the lesion is analyzed with a targeted gene panel on an FNA specimen and the diagnosis is indeterminate for malignancy (Bethesda III or IV), the patient may not undergo surgical resection, as was reported recently for a series of 178 patients who had Bethesda III/IV lesions and no mutations identified on molecular testing [58]. As a result, our data make a case for the inclusion of *FLCN* gene coverage on such commercial assays. Even in cases with *FLCN* homozygous deletion, presumably representing a secondary alteration, primary canonical driver mutations were not identified in four cases from the database subsets, meaning that these cases have the potential to be misclassified as benign depending on the methodology of the assay. To further underscore the importance of the inclusion of the *FLCN* gene in these assays, we are likely underestimating the number of thyroid tumors with *FLCN* alterations. The frequency reported in our study was on a cohort of over 3000 patients with thyroid carcinoma and did not include patients with benign neoplasms such as follicular or oncocytic adenomas. Even if the frequency is assumed to be 1.1%, as reported in our dataset analyses, that may not be an insignificant number of patients given the frequency in which preoperative molecular assays are used to determine patient selection for surgery.

## 5. Conclusions

In summary, *FLCN* alterations were initially recognized in patients with BHD syndrome, typically leading to oncocytic renal tumors, fibrofolliculomas, and lung cysts. More recently, however, a broader range of tumors have been identified in BHD patients, and numerous other tumors have been found to harbor somatic *FLCN* alterations. Until this study, however, there have been only isolated reports of thyroid tumors in patients with BHD, and no studies to date have documented somatic *FLCN* alterations in thyroid tumors. Our findings suggest that *FLCN* mutations, though rare, may represent an early oncogenic driver in thyroid neoplasia, detectable in both germline and somatic settings and often leading to oncocytic features. In contrast, secondary *FLCN* alterations are more commonly seen in aggressive thyroid carcinomas, where they may not result in an oncocytic histology. Additionally, the lack of *FLCN* gene coverage in standard thyroid molecular panels likely contributes to its under-reporting in clinical practice. This under-detection is notable, as we identified four malignant cases with *FLCN* homozygous deletions and no other detectable oncogenic mutations by RNA or DNA sequencing. Broadening genetic panels to include *FLCN* could improve the detection and characterization of *FLCN*-altered thyroid tumors. Furthermore, review of additional *FLCN*-mutated thyroid tumors is essential for further characterization.

## Figures and Tables

**Figure 1 curroncol-32-00224-f001:**
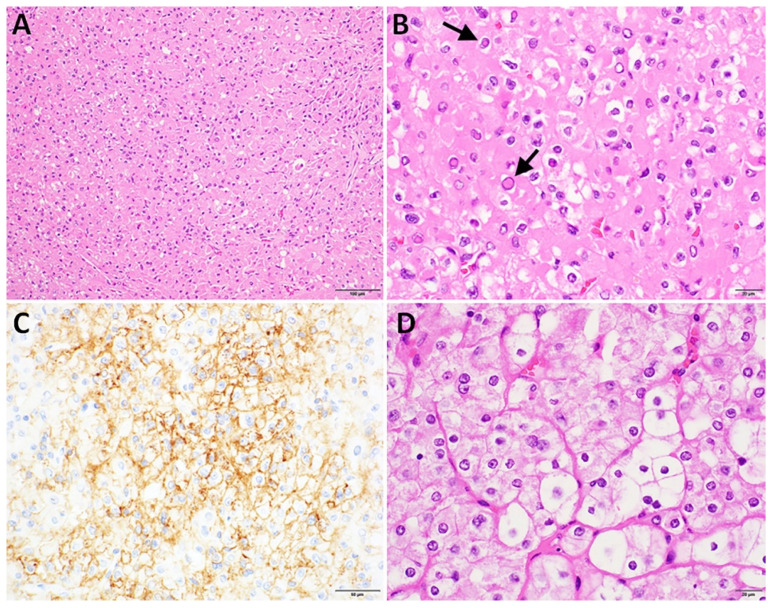
Patient 1 oncocytic adenoma. (**A**) Diffuse solid growth of oncocytic cells without colloid (hematoxylin and eosin stain, 200× magnification, and 100 micron scale bar). (**B**) Thyroid tumor cell nuclei mimic nuclei of papillary thyroid carcinoma including many cells with bright, eosinophilic nuclear pseudo-inclusions (arrows), cells with ‘orphan annie’ nuclear clearing and cells with crumpled, ‘raisinoid’ nuclei (hematoxylin and eosin, 600× magnification, and 20 micron scale bar). (**C**) Thyroid tumor shows patchy membranous expression of CD56 (CD56-immunohistochemical stain with hematoxylin counterstain, 400× magnification, and 50 micron scale bar). (**D**) Patient’s prior kidney tumor shows features consistent with chromophobe carcinoma (hematoxylin and eosin, 600× magnification, 20 micron scale bar).

**Figure 2 curroncol-32-00224-f002:**
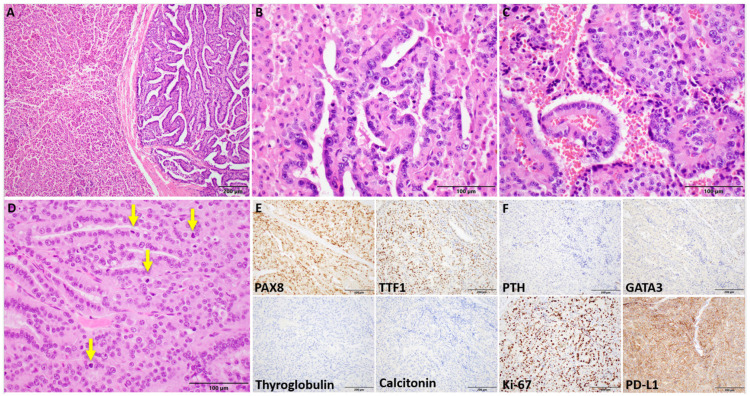
Patient 2. Oncocytic poorly differentiated thyroid carcinoma. (**A**) Sharp demarcation between solid (left) and papillary-like (right) architecture (100× magnification and 100 micron scale bar). (**B**) Papillary-like architectural growth pattern arising from a solid growth pattern background (400× magnification and 100 micron scale bar). (**C**) Papillary-like frond and adjacent solid growth pattern, both exhibiting oncocytic cytoplasm with an absence of papillary thyroid carcinoma (PTC) nuclear features (400× magnification and 100 micron scale bar). (**D**) Multiple mitotic figures (arrows), consistent with a poorly differentiated thyroid carcinoma (400× magnification and 100 micron scale bar). ((**A**–**D**) hematoxylin and eosin-stained tumor sections). (**E**,**F**) Immunohistochemical expression profile (200× magnification and 200 micron scale bar).

**Figure 3 curroncol-32-00224-f003:**
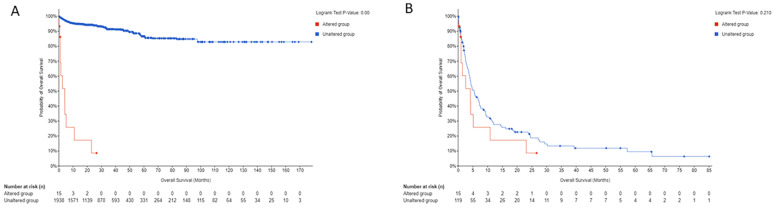
(**A**) Overall survival was compared between patients with *FLCN* driver alterations versus those without across all public datasets included in this study when survival data were available. Patients with *FLCN* driver alterations exhibited significantly worse overall survival. (**B**) Given that 24 of the 25 patients with *FLCN* driver alterations in the public datasets included in this study had anaplastic thyroid carcinoma, a subgroup analysis was conducted to evaluate the impact of *FLCN* on overall survival in this aggressive subset. While there was a trend toward poorer survival in patients with *FLCN* alterations, the difference did not reach statistical significance.

**Table 1 curroncol-32-00224-t001:** Thyroid carcinomas with *FLCN* alterations in the Orien-Avatar database.

Age at Dx	Sex	Pathologic Dx	*FLCN* Gene Coverage	Somatic vs. Germline *FLCN* Alteration	*FLCN* Alteration	VAF	Variant Reads	Reference Reads	Other Mutations	Family Cancer History	Overall Survival (Months)	Vital Status	T Stage at Dx	N Stage at Dx	Metastases
71.8	M	Oncocytic carcinoma	Germline status only, *FLCN* not covered on tumor NGS panel	Germline	G319fs	0.42	154	214	Tumor sequenced (*FLCN* gene not on panel)	“Colon, NOS”	25	Living	T3	N1b	Liver, lung, rib, sternum, clavicle and associated joints, and adrenal gland; lymph nodes of head, face, and neck
43	F	PTC, Follicular Variant	Germline and Tumor	Somatic	Biallelic loss	N/A	N/A	N/A	*BRAF* V600E (known oncogenic; likely GOF), *ARID1A* E1735* (likely oncogenic; likely LOF)	N/A	59	Living	T3	N0	N/A
46	M	PTC, NOS	Germline and Tumor	Somatic	Biallelic loss	N/A	N/A	N/A	*BRAF* V600E (known oncogenic; likely GOF)	Lung NOS, Thyroid gland	62	Living	T1a	N1a	Rib, sternum, and clavicle and associated joints; lymph nodes of head, face, and neck
39	F	PTC, NOS	Germline and Tumor	Somatic	Biallelic loss	N/A	N/A	N/A	No oncogenic point mutations	N/A	113	Living	T3	N0	N/A
72	M	PTC, NOS	Germline and Tumor	Somatic	Biallelic loss in metastasis but not in primary	N/A	N/A	N/A	Primary and lymph node metastasis: *BRAF* V600E (known oncogenic; likely GOF)	Kidney, NOS, Skin, NOS, Thyroid gland	51	Living	T1b	N1b	Lymph nodes of head, face, and neck
14	F	PTC, NOS	Germline and Tumor	Somatic	Biallelic loss	N/A	N/A	N/A	*BRAF* V600E (known oncogenic; likely GOF)	N/A	80	Living	T3	N1a	Lymph nodes of head, face, and neck
44	F	PTC, NOS	Germline and Tumor	Somatic	Biallelic loss	N/A	N/A	N/A	*BRAF* V600E (known oncogenic; likely GOF)	Skin, NOS	84	Living	T1b	N1b	Lymph nodes of head, face, and neck
67	M	PTC, NOS	Germline and Tumor	Somatic	Biallelic loss	N/A	N/A	N/A	*BRAF* V600E (known oncogenic; likely GOF)	N/A	42	Living	T3	N1b	Lymph nodes of head, face, and neck
41	F	PTC, NOS	Germline and Tumor	Somatic	Biallelic loss in metastasis but not in primary	N/A	N/A	N/A	Primary and lymph node metastasis: *BRAF* V600E (known oncogenic; likely GOF)	Thyroid Gland	75	Living	T3	N1b	Lymph nodes of head, face, and neck
23	F	PTC, NOS	Germline and Tumor	Somatic	Biallelic loss	N/A	N/A	N/A	*BRAF* V600E (known oncogenic; likely GOF)	N/A	20	Living	T2	N1a	Lymph nodes of head, face, and neck

Legend: DX, diagnosis; *FLCN*, folliculin; VAF, variant allele frequency; T, tumor; N, nodal; M, male; F, female; NOS, not otherwise specified; N/A, not applicable; GOF, gain of function; LOF, loss of function.

**Table 2 curroncol-32-00224-t002:** Thyroid carcinomas with *FLCN* alterations in the CBioPortal database.

Database	Age at Dx	Sex	Pathologic Dx	*FLCN* Gene Coverage	*FLCN* Alteration	VAF	Variant Reads	Reference Reads	Other Mutations	Overall Survival (Months)	Vital Status	T Stage at Dx	Metastasis
Anaplastic Thyroid Cancers (GATCI, Cell Reports 2024)	<70	M	ATC	Tumor	Q220* truncating mutation	0.54	22	41	*BRAF* V600E (oncogenic; gain of function), *TP53* M246I (likely oncogenic; likely loss of function), *NF2* Q453* (likely oncogenic; likely loss of function)	4	Deceased	T4b	Yes (Nodal + Distal)
Anaplastic Thyroid Cancers (GATCI, Cell Reports 2024)	>=70	F	ATC	Tumor	Biallelic loss	N/A	N/A	N/A	*TP53* K132N; likely oncogenic; likely loss of function	1	Deceased	T4b	Yes (Nodal + Distal)
Anaplastic Thyroid Cancers (GATCI, Cell Reports 2024)	>=70	F	ATC arising in a background of PTC	Tumor	Biallelic loss	N/A	N/A	N/A	*BRAF* V600E (oncogenic; gain of function)	4	Deceased	T4b	Yes (Nodal + Distal)
Anaplastic Thyroid Cancers (GATCI, Cell Reports 2024)	<70	M	ATC	Tumor	Biallelic loss	N/A	N/A	N/A	*BRAF* V600E (oncogenic; gain of function), *ASXL2* Q1013 (likely oncogenic; likely loss of function)	<1	Deceased	T4b	Yes (Nodal + Distal)
Anaplastic Thyroid Cancers (GATCI, Cell Reports 2024)	>=70	M	ATC arising in a background of PTC	Tumor	Biallelic loss	N/A	N/A	N/A	*BRAF* V600E (oncogenic; gain of function), *TP53* M246I (likely oncogenic; likely loss of function)	3	Deceased	T4b	Yes (Nodal + Distal)
Anaplastic Thyroid Cancers (GATCI, Cell Reports 2024)	<70	M	ATC	Tumor	Biallelic loss	N/A	N/A	N/A	*TP53* Q331H; likely oncogenic; likely loss of function	23	Deceased	T4b	N/A
Anaplastic Thyroid Cancers (GATCI, Cell Reports 2024)	Not provided	F	ATC	Tumor	Biallelic loss	N/A	N/A	N/A	No oncogenic point mutations	N/A	N/A	N/A	N/A
Anaplastic Thyroid Cancers (GATCI, Cell Reports 2024)	>=70	M	ATC	Tumor	Biallelic loss	N/A	N/A	N/A	*NRAS* Q61R (oncogenic gain of function), *TP53* C135W (likely oncogenic; likely loss of function)	3	Deceased	T4b	Yes (Nodal + Distal)
Anaplastic Thyroid Cancers (GATCI, Cell Reports 2024)	>=70	F	ATC	Tumor	Biallelic loss	N/A	N/A	N/A	*NF2* E186* (likely oncogenic; likely loss of function)	11	Deceased	N/A	Yes (Distal)
Anaplastic Thyroid Cancers (GATCI, Cell Reports 2024)	>=70	F	ATC	Tumor	Biallelic loss	N/A	N/A	N/A	*BRAF* V600E (oncogenic; gain of function)	1	Deceased	N/A	No
Anaplastic Thyroid Cancers (GATCI, Cell Reports 2024)	<70	M	ATC	Tumor	Biallelic loss	N/A	N/A	N/A	*STK11* Q220* (likely oncogenic; likely loss of function)	1	Deceased	T4b	Yes (Nodal)
Anaplastic Thyroid Cancers (GATCI, Cell Reports 2024)	>=70	F	ATC	Tumor	Biallelic loss	N/A	N/A	N/A	*NRAS* Q61K (oncogenic; gain of function) + *PIK3CA* Q546L (likely oncogenic; likely gain of function)	1	Deceased	N/A	No
Anaplastic Thyroid Cancers (GATCI, Cell Reports 2024)	>=70	M	ATC arising in a background of onocytic carcinoma	Tumor	Biallelic loss	N/A	N/A	N/A	*ATR* X878_splice in anaplastic (splice; likely oncogenic; likely loss of function); *EP400* X2778_splice in oncocytic carcinoma (splice; likely oncogenic; likely loss of function)	1	Deceased	N/A	No
Anaplastic Thyroid Cancers (GATCI, Cell Reports 2024)	>=70	F	ATC arising in a background of PTC	Tumor	Biallelic loss	N/A	N/A	N/A	PTC component only: *BRAF* V600E (oncogenic; gain of function), *TP53* E285K (likely oncogenic; likely loss of function), *BTG1* Q82* (likely oncogenic; likely loss of function)	3	Living	N/A	N/A
Anaplastic Thyroid Cancers (GATCI, Cell Reports 2024)	<70	M	ATC arising in a background of oncocytic carcinoma	Tumor	Biallelic loss	N/A	N/A	N/A	Oncocytic carcinoma component only: *PTEN* W274* (likely oncogenic; likely loss of function)	6	Deceased	N/A	No
Anaplastic Thyroid Cancers (GATCI, Cell Reports 2024)	<70	M	ATC arising in a background of poorly differentiated thyroid carcinoma	Tumor	Biallelic loss	N/A	N/A	N/A	No oncogenic point mutations	19	Deceased	N/A	No
Anaplastic Thyroid Cancers (GATCI, Cell Reports 2024)	<70	F	ATC	Tumor	Biallelic loss	N/A	N/A	N/A	*NRAS* Q61R (oncogenic + gain of function), *TP53* I195T (likely oncogenic; likely loss of function)*, EIF1AX* G9R (likely oncogenic; likely gain of function)	1	Deceased	N/A	No
Anaplastic Thyroid Cancers (GATCI, Cell Reports 2024)	<70	M	ATC	Tumor	Biallelic loss	N/A	N/A	N/A	*NRAS* Q61R (oncogenic + gain of function)	<1	Living	N/A	No
Anaplastic Thyroid Cancers (GATCI, Cell Reports 2024)	>=70	M	ATC	Tumor	Biallelic loss	N/A	N/A	N/A	*BRAF* V600E (oncogenic + gain of function), *PTEN* G129R (oncogenic + gain of function), *PICK3CA* E453K (oncogenic + gain of function), *CHEK2* I157T (oncogenic + gain of function), *MUTHY R*233* (likely oncogenic; likely loss of function)	5	Deceased	T4b	Yes (Nodal + Distal)
Anaplastic Thyroid Cancers (GATCI, Cell Reports 2024)	>=70	M	ATC likely arising out of a follicular carcinoma	Tumor	Biallelic loss	N/A	N/A	N/A	*NRAS* Q61R (oncogenic + gain of function)	1	Living	T4a	Yes (Nodal)
Anaplastic Thyroid Cancers (GATCI, Cell Reports 2024)	>=70	F	ATC	Tumor	Biallelic loss	N/A	N/A	N/A	*BRAF* V600E (oncogenic + gain of function), *RAD51C P21A* (likely oncogenic + known biological effect), *NF2* Q178* (likely oncogenic; likely loss of function)	27	Living	T4b	Yes (Nodal)
Anaplastic Thyroid Cancers (GATCI, Cell Reports 2024)	>=70	F	ATC arising out of a PTC	Tumor	Biallelic loss	N/A	N/A	N/A	No oncogenic point mutations	3	Deceased	T4a	Yes (Nodal)
Anaplastic Thyroid Cancers (GATCI, Cell Reports 2024)	>=70	M	ATC arising out of a PTC	Tumor	Biallelic loss	N/A	N/A	N/A	*TP53* R273C (likely oncogenic; loss of function) in both anaplastic and PTC components	5	Deceased	T4a	Yes (Nodal)
Anaplastic Thyroid Cancers (GATCI, Cell Reports 2024)	<70	M	ATC arising out of a follicular carcinoma	Tumor	Biallelic loss	N/A	N/A	N/A	*NRAS* G13R in anaplastic and PTC components (oncogenic + gain of function), *EIF1AX* X113_splice in follicular carcinoma component only (likely oncogenic + likely gain of function)	3	Deceased	T4b	Yes (Distal)
Thyroid Carcinoma (TCGA, PanCancer Atlas)	36	F	PTC with predominantly follicular architecture with oncocytic features and focal tall cell features	Tumor	Biallelic loss	N/A	N/A	N/A	Not profiled for point mutations	40	Living	T3	Yes (Distal, Lung)

Legend: Dx = diagnosis; M, male; F, female; *FLCN* = folliculin; VAF = variant allele frequency; T, tumor; N/A, not applicable.

## Data Availability

The original contributions presented in this study are included in this article. Data mined from publicly available datasets can be retrieved from interrogating the referenced datasets at “www.cBioPortal.org (last accessed on 3 June 2024). Data from the Orien-Avatar Database are not publicly available and relevant data are presented in this article. Further inquiries can be directed to the corresponding author.

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
