# Peer review of "Folliculin* (*FLCN*) in Thyroid Tumors: Incidence, Significance, and Role as a Driver Gene and Secondary Alteration"

_curroncol, 2025, doi:10.3390/curroncol32040224_

Round 1
Reviewer 1 Report
Comments and Suggestions for Authors
In this article, Hassan et al. discuss the role of FLNC in thyroid tumors.
The topic is interesting, all the more so since it is concluded that the evaluation of mutations in this gene should be included in routine practices. At the same time there are some points that should be reviewed.
In particular:
line 76: Authors should explain on what type of biological sample and what method and include a literature reference under which the NGS analysis was performed.
line 122: As for Patient 2, the diagnosis and surgery were performed at another hospital?
line 149: "BHD sindrome" this information should already be introduced into patient classification in lines 122-123..
Table 1 is not very clear, maybe, it should be rethought and maybe, it would be better to have it all on one page because the reader might get confused when reading it.
line 242-243: so if it is unknown whether there are somatic mutations in the FLCN gene. How do you determine its impact on the tumor?
Table 2: the same observatin as in table 1.
line 415-416: this concept has already been introduced and is repeated again.
In general, the discussion requires a thorough revision, as it exhibits a repetitive structure in several sections, with identical cases being reiterated multiple times. This redundancy not only affects the coherence of the discourse but also risks causing confusion for the reader, making it challenging to follow the logical flow of the argument.
The article needs a major revision
Author Response
Responses to Reviewer #1
Comment 1: line 76: Authors should explain on what type of biological sample and what method and include a literature reference under which the NGS analysis was performed.
Response 1: Thank you for your comment and allowing us to further clarify the methodology. Please see new lines 80-84. “For each patient, we conducted a thorough review of all available clinical and laboratory data, including molecular sequencing results from the Afirma® Genomic Sequencing Classifier, performed on aspirated thyroid samples, as well as MI Profile Testing through Caris® Life Sciences, conducted on formalin-fixed, paraffin-embedded tissue.” Two citations have been added that describe the sequencing methodology, include one for the Afirma NGS test and another for the Caris NGS test.
Comment 2: line 122: As for Patient 2, the diagnosis and surgery were performed at another hospital?
Response 2: Yes, the patient initially presented to an outside hospital and underwent a right thyroid lobectomy. As we mention in the next paragraph, the patient was subsequently transferred to our institution for further management. As is customary when a patient transfers care to our comprehensive cancer center, all outside pathology slides are sent to us for review prior to further therapeutic intervention for confirmation of the diagnosis when possible. In this case, the patient’s slides were reviewed and are described in the manuscript. After our review, the patient underwent a completion thyroidectomy, with resection of additional masses as we describe in the manuscript. This is all part of the normal workflow as a tertiary care center. Does that answer your question?
Comment 3: line 149: "BHD sindrome" this information should already be introduced into patient classification in lines 122-123..
Response 3: Thank you for clarifying. We have updated the manuscript to explicitly state that the patient carries a diagnosis of BHD syndrome in new lines 127. It now reads as follows: “A 44-year-old female with BHD syndrome (confirmed FLCN germline mutation) and a history of multifocal chromophobe renal cell carcinoma presented with a newly detected thyroid nodule on routine physical exam.”
Comment 4: Table 1 is not very clear, maybe, it should be rethought and maybe, it would be better to have it all on one page because the reader might get confused when reading it.
Response 4: Thank you for pointing this out. We have reformatted Table 1 to make the headings and the content more readable. We will work with the editing team if this paper gets accepted to ensure adequate formatting that is adherent to journal standards.
Comment 5: line 242-243: so if it is unknown whether there are somatic mutations in the FLCN gene. How do you determine its impact on the tumor?
Response 5: Thank you for your comment. We present a total of 37 patients with FLCN alterations. Your reference is to one of these patients, for which we address that we cannot fully describe the role of this gene because we do not know if there is an additional somatic alteration. We state, “Unfortunately, the patient with the germline mutation did not have FLCN gene coverage on their somatic tumor NGS panel, meaning the VAF is unknown. However, the absence of other alterations may indicate that it plays a significant role. However, since that would be conjecture, our discussion and conclusion focuses on the cases for which we do have this information, which includes the other 36 patients identified.
Comment 6: Table 2: the same observatin as in table 1.
Response 6: Thank you. We have reformatted Table 2 as well. Please see response 4 above.
Comment 6: line 415-416: this concept has already been introduced and is repeated again.
Response 6: Thank you for your comment. The discussion has been revised to decrease the repetitive and reiterative structure you mention in comment 7.The section you reference has been removed.
Comment 7: In general, the discussion requires a thorough revision, as it exhibits a repetitive structure in several sections, with identical cases being reiterated multiple times. This redundancy not only affects the coherence of the discourse but also risks causing confusion for the reader, making it challenging to follow the logical flow of the argument.
Response 7: Thank you for your observation. We have made changes to the discussion to streamline it for readability and to remove redundancy.
Reviewer 2 Report
Comments and Suggestions for Authors
The article titled “Folliculin (FLCN) in Thyroid Tumors: Incidence, Significance, and Role as a Driver Gene and Secondary Alteration” used two clinical cases and public database to discuss the FLCN gene alteration in thyroid tumors. This article is informative and well written; however, the clinical sample size is too small to draw accurate conclusions. The information in the public database in this article can provide some correlation between FLCN mutations and thyroid cancer. Here are some detailed comments:
- The authors italicized the FLCN gene name, which is correct, but some formatting was missed, such as line 48, line 39 and line 123, line 152, line155, line227, please check the full text carefully.
- In the introduction section, please briefly explain the difference between FLCN germline mutation and somatic mutation.
- Based on what clinical features was the first patient diagnosed with BDH syndrome? Is BDH syndrome caused by the FLCN mutation?
- The scale bar marking in Figure 1 is very unclear. Please keep the scale line and note the scale value in the figure legend.
- It looks weird to use pictures with different magnifications in the same set of pictures. Please change the magnification of the four pictures in Figure 1 to the same.
- Figure 2 has the same problem and needs to be scaled.
- In line 173, what does “FNA” mean? If you use an abbreviation, use the full name the first time it appears in the text.
- Based on the clinical data of the two patients, there is no direct evidence to prove that their thyroid tumors were caused by FLCN gene mutations. What explanation does the author have for this?
- The first patient had no recurrence symptoms after surgery, while the second patient had a series of complications after surgery. Please discuss whether this may be related to the different mutation sites of FLCN.
- Table 1 and table 2,
- Please italicize the FLCN
- Please adjust the font size and layout of Tables to make it clearer.
Author Response
Responses to Reviewer 2
Comment 1: The authors italicized the FLCN gene name, which is correct, but some formatting was missed, such as line 48, line 39 and line 123, line 152, line155, line227, please check the full text carefully.
Response 1: Thank you for your attention to detail. It also appears as though some of the formatting in the converted PDF is different than the word document. We have gone through the manuscript and have updated the italicized areas when referring to the gene and have made sure to keep it regularly formatted when referring to the FLCN protein.
Comment 2: In the introduction section, please briefly explain the difference between FLCN germline mutation and somatic mutation.
Response 2: Thank you. We have added a parenthetical statement after germline with “hereditary” and after somatic with “acquired”. Going into this further is likely not necessary for anyone who would be reading Current Oncology, unless you have a more specific comment/reason for expanding upon this in the introduction.
Comment 3: Based on what clinical features was the first patient diagnosed with BDH syndrome? Is BDH syndrome caused by the FLCN mutation?
Response 3: The patient was diagnosed with BHD syndrome via the major criterion of a pathogenic germline FLCN mutation as well two minor criterion (early onset <50 years of age renal cell carcinoma and multifocal renal carcinoma). A pathogenic FLCN mutation is a major diagnostic criterion and is pathognomonic for BHD syndrome. I have attempted to add this without distracting the flow the paper.
Comment 4: The scale bar marking in Figure 1 is very unclear. Please keep the scale line and note the scale value in the figure legend.
Response 4: Thank you for bringing it to our attention. We have kept the scale bar and have noted the annotation in the legend. For additional details, we have included the magnification of these photos in the caption.
Comment 5: It looks weird to use pictures with different magnifications in the same set of pictures. Please change the magnification of the four pictures in Figure 1 to the same.
Response 5: Different magnifications are used to highlight different features, such as the diffuse growth pattern in A, as well as the patchy staining seen in C, which are important features to highlight and cannot be appreciated at higher magnification. Taking the other photos at lower magnification would not allow for the appreciation of the features we are highlighting. This is customary when presenting histologic and immunohistochemical images, and, as pathologists, we understand the importance.
Comment 6: Figure 2 has the same problem and needs to be scaled.
Response 6: Thank you. The scaling issue has been fixed. Please see updated Figure 2. Captions also now include scale bar notations.
Comment 7: In line 173, what does “FNA” mean? If you use an abbreviation, use the full name the first time it appears in the text.
Response 7: This was already done on line 123 when the phrase FNA was first used.
Comment 8: Based on the clinical data of the two patients, there is no direct evidence to prove that their thyroid tumors were caused by FLCN gene mutations. What explanation does the author have for this?
Response 8: This is not the conclusion that we are drawing. While direct causality between FLCN gene mutations and the development of thyroid tumors would require transgenic animal models, the paper highlights several points that suggest its role in patients with thyroid neoplasms. The first patient, with a germline FLCN mutation and a history of BHD syndrome. Importantly, the tumor lacked other oncogenic mutations, which suggests that FLCN may act as an early neoplastic driver. The second patient had a bilallelic inactivating mutation, as well as when combining the broader analysis of thyroid carcinoma cases from sequencing databases, indicate that while FLCN alterations may play a role in early tumorigenesis, especially in cases with oncocytic features and in progression, especially in cases with advanced features. Either way, the presence of FLCN mutations in both germline and somatic settings, particularly in oncocytic tumors, provides strong evidence for further investigation into FLCN’s role in thyroid neoplasms. I’d point the discussion and conclusion for a more thorough response. We’ve been very careful not to overstate findings and to be thoughtful about our discussion and conclusion.
Comment 9: The first patient had no recurrence symptoms after surgery, while the second patient had a series of complications after surgery. Please discuss whether this may be related to the different mutation sites of FLCN.
Response 9: As we mention in the discussion and throughout the text, the difference is likely due to the co-existing TP53 and RB1 mutations, along with the TERT promoter mutation, which are well-known to be associated with high-grade transformation in thyroid carcinomas. The larger database analyses also support this sequence in tumor behavior, as is described in the discussion as well. Please let me know if you do not think this is clear.
Comment 10: Table 1 and table 2, Please italicize the FLCN
Response 10: Thank you for picking this up. We have updated the italics.
Comment 11: Please adjust the font size and layout of Tables to make it clearer.
Response: 11: We have updated the Tables and will work closely with the formatting team to abide by journal specifications if accepted for publication.
Round 2
Reviewer 1 Report
Comments and Suggestions for Authors
I would like to thank the Authors for their thoroughness in making the suggested changes. In particular, the discussion now appears smoother and less repetitive.
Reviewer 2 Report
Comments and Suggestions for Authors
The authors addressed all my comments.